# Flooding and elevated prenatal depression in rural Bangladesh: A mixed methods study

Suhi Hanif[1], Jannat-E-Tajreen Momo[2], Farjana Jahan[2], Liza Goldberg[3],
Natalie Herbert[3], Afsana Yeamin[2], Abul Kasham Shoab[2], Reza Mostary Akhter[4],
Sajal Kumar Roy[5], Gabriella Barratt Heitmann[1], Ayse Ercumen[6], Mahbubur Rahman[2,7],
Fahmida Tofail[4], Gabrielle Wong-Parodi[3,8,9], Jade Benjamin-Chung[1,10]*

1 Department of Epidemiology and Population Health, Stanford University, Stanford, California, United
States of America, 2 Environmental Health and WASH, International Centre for Diarrhoeal Disease
Research, Dhaka, Bangladesh, 3 Department of Earth System Science, Stanford University, Stanford,
California, United States of America, 4 Maternal and Child Nutrition, International Centre for Diarrhoeal
Disease Research, Dhaka, Bangladesh, 5 Bangladesh Water Development Board, Dhaka, Bangladesh,
6 College of Natural Resources, North Carolina State University, Raleigh, North Carolina, United States
of America, 7 Global Health and Migration Unit, Department of Women's and Children's Health, Uppsala
University, Uppsala, Sweden, 8 Department of Environmental Social Sciences, Stanford University,
Stanford, California, United States of America, 9 Woods Institute for the Environment, Stanford University,
Stanford, California, United States of America, 10 Chan Zuckerberg Biohub, San Francisco, California,
United States of America

* jadebc@stanford.edu

Science, UNITED STATES OF AMERICA

**Peer Review History:** PLOS recognizes the
benefits of transparency in the peer review
process; therefore, we enable the publication
of all of the content of peer review and
author responses alongside final, published
articles. The editorial history of this article is
available here: https://doi.org/10.1371/journal.
pgph.0004792

## Abstract

Prenatal depression can have lasting adverse impacts on child health. Little is known
about the impact of floods on prenatal depression in low- and middle-income countries. We conducted a cross-sectional survey of 881 pregnant women from September 24, 2023 to July 19, 2024 in riverine communities in rural Bangladesh. We
recorded participant-reported flooding in the past 6 months, administered the Edinburgh Postnatal Depression Scale (EPDS), and obtained water level data and remote
sensing data on distance to surface water. We fit generalized linear and log-linear
models adjusting for month, wealth, education, age, and gestational age. We conducted 2 focus group discussions with 20 adult women. 3.6% of compounds were
flooded in the past 6 months. Flooding of compounds was associated with elevated
depression (adjusted prevalence ratio (aPR) = 2.08, 95% CI 1.24, 3.51) and thoughts
of self-harm (aPR = 8.40, 95% CI 4.19, 16.10). Latrine flooding was associated with
higher depression (aPR = 3.58, 95% CI 2.22, 5.75)). Higher water levels and shorter
distance to permanent surface water were significantly associated with mean EPDS
scores. Focus groups revealed that domestic violence, inadequate sanitation, gendered vulnerabilities in accessing latrines, childcare difficulties, and food insecurity
were key drivers of depression due to floods. Flood preparedness strategies included
relocation, storing food, and home modifications. In summary, in rural Bangladesh,
flooding, higher water levels, and proximity to water bodies were associated with

**Data availability statement:** Data is published in Open Science Framework, DOI 10.17605/OSF.IO/VW9BJ.

**Funding:** This work was supported by the National Institute of Child Health and Human Development (R01HD108196 to JBC). The funders had no role in study design, data collection and analysis, decision to publish, or preparation of the manuscript. The content is solely the responsibility of the authors and does not necessarily represent the official views of the National Institutes of Health. JBC is a Chan Zuckerberg Biohub Investigator. icddr,b acknowledges with gratitude the commitment of Grand Challenges Canada to its research efforts. icddr,b is also grateful to the Governments of Bangladesh and Canada for providing core/unrestricted support. Research was also supported by grants to GWP from the Stanford King Center on Global Development and the Department of Earth System Science at the Doerr School of Sustainability at Stanford University.

**Competing interests:** The authors have declared that no competing interests exist.

prenatal depression, and depression following floods was strongly influenced by inadequate sanitation and hygiene infrastructure.

## Introduction

Flooding can have negative impacts on mental health outcomes [1,2]. This is particularly the case in low- and middle-income countries (LMICs), which are more vulnerable to climate hazards. In South Asia, flooding is projected to increase under climate change [3], and it is associated with depression, anxiety, and post-traumatic stress disorder [4].

Pregnant women face higher risks of mental health conditions including depression and self-harm [5]. Experiencing flooding during pregnancy can result in peritraumatic distress – distress experienced within the period following a traumatic event – and has been shown to increase the risk of prenatal depression in high income settings [6,7]. However, no prior studies have investigated the impacts of flooding on prenatal mental health in LMICs, [8] where 26% of women experience perinatal depression [9]. Flooding in LMICs could influence prenatal mental health through multiple pathways including reduced agricultural productivity; lower income; damage to household infrastructure; loss of possessions; power outages; lack of access to safe water, sanitation, and hygiene; displacement; trauma; death of family members; disrupted communication systems; and interruptions to prenatal or obstetric care and other health system [4,10–12]. In some populations, gender norms require women to stay within the home unless accompanied by men and to keep their clothes dry can make the process of coping with floods more difficult for women, and their typical responsibility to serve as children's primary caregivers can impede their efforts to adequately prepare for and remain safe during floods [13].

Prenatal depression is associated with adverse outcomes such as preterm births [14,15], low birth weight, and impaired child development [16,17]. Exposure to stress and excess glucocorticoids in utero may result in epigenetic changes and immunologic, metabolic, neuroendocrine, and cognitive impacts associated with illnesses over the life course [18]. Given the potential long-lasting effects of in utero stressors on children over their lifetimes, it is essential to understand the impacts of flooding on the mental health of populations most vulnerable to climate change.

At the 2023 COP28 UN Climate Change Conference, countries agreed to a framework for the Global Goal on Adaptation with the goal of enhancing "adaptive capacity, strengthening resilience and reducing vulnerability to climate change" [19]. The framework lays out areas that will require adaptation interventions, including health, but no measurable indicators of specific strategies have been identified, in part due to a need for more evidence. Recent reviews of adaptation strategies for mental health globally and for public health in LMICs both concluded that there is a dearth of evidence on this topic [20,21]. For example, few studies have investigated individual- or household-level adaptation strategies to flooding in LMICs [22].

Bangladesh ranks as the world's 7th most climate-affected country [23]. Its unique hydrogeology places the majority of the population at flood risk: 75% of the country

is within 10 meters above sea level and floodplains constitute 80% of its land mass [10]. The seasonal monsoon results in extreme precipitation and frequent flooding in Bangladesh. Forty-five percent of the total population of Bangladesh is at risk of riverine flooding, which is the highest in the world [24], and climate change is predicted to increase the frequency of floods in the future [3]. The majority of the population living in floodplains is rural and poor [25]. Approximately 4% of the population lives on riverine islands known as chars, where flooding and erosion are even more common [26]. Eighty percent of char residents are ultra-poor, and most do not own land, lack access to healthcare and education, and are excluded from state services [27,28]. Another study in northeastern Bangladesh on the spatial distribution of mental health outcomes in adults after riverine flooding found participants with moderate major depressive symptoms lived adjacent to the river [29]. However, no prior studies in climate-sensitive populations in Bangladesh have investigated the influence of flooding on prenatal depression.

In this mixed methods study, including a cross-sectional survey and focus group discussions, our objective was to investigate the association between flooding and prenatal depression in a climate-vulnerable population residing in chars and riverine communities in rural Bangladesh. Additionally, we sought to understand potential drivers of depression following flooding and to identify household strategies for preparing and responding to floods to inform future adaptation interventions for climate resilience. Our overarching aim was to generate evidence to inform climate adaptation strategies that promote women's mental health, and in turn, healthy births, and child growth and development.

## Methods

### Study design

We used a mixed method approach, triangulating findings from a quantitative cross-sectional baseline survey from a randomized trial and focus group discussions. The quantitative survey was conducted in a large sample of 881 pregnant women, which allowed us to estimate associations between flooding, surface water proximity, and water levels and prenatal depression. The survey also characterized adaptation practices. To further elicit potential mechanisms through which flooding could contribute to prenatal depression, we conducted focus group discussions (FGDs). This approach allowed us to characterize flood-related vulnerabilities and adaptation strategies through in-depth, nuanced discussions and personal narratives that may not be captured in a standardized survey.

### Quantitative survey

This analysis used data from the baseline survey of the Cement-based flooRs AnD chiLd hEalth (CRADLE) trial, an individually randomized trial in Sirajganj and Tangail districts, Bangladesh that will test whether replacing soil floors with concrete floors improves maternal and child health (NCT05372068) [30]. This study site was chosen because it has a higher prevalence of soil-transmitted helminth infections (the trial's primary outcome) compared to other regions of Bangladesh [31]. The recruitment period for the trial was between August 8, 2023 and July 17, 2024. The trial randomized study households to intervention or control with a one-to-one ratio within geographic blocks of 10 contiguous households. A minimum 100 meter buffer zone was enforced between all study households to minimize contamination. The baseline survey was conducted in 881 households; 81 of these were excluded from the main trial during the formation of geographic blocks, but we included them in this analysis to maximize the sample size.

Households in our study were located in a flood-prone region adjacent to the Jamuna River in northeastern Bangladesh. The study population includes vulnerable communities living on chars, who are at a particularly high risk of flooding. The cross-sectional baseline survey was administered to pregnant women enrolled in the trial. The eligibility criteria for enrollment in the trial were as follows: households with a woman in her second or third trimester of pregnancy, household floor constructed of soil, household wall not made of earth, and no plan to relocate for 3 years. The survey included questions about flood history and flood preparation practices, a module on maternal depression, as well as household demographic characteristics, maternal education, assets, animal husbandry, and child illness.

## Exposures

We measured self-reported flooding with a recall period of 6 months. Participants were asked to report whether floods occurred within their union (the smallest rural administrative unit in the local government), their compound (typically 2–4 households of blood relatives that share latrines and food), or their latrine. We also asked participants to report approximately how many months ago flooding occurred and for how many days their compound or latrine was flooded. Additionally, we obtained data on observed water levels in meters above mean sea level (mMSL) for Sirajganj District from the Bangladesh Water Development Board Flood Forecasting and Warning Centre. Water level was recorded daily at three-hour intervals by a designated gauge reader from the Bangladesh Water Development Board. We calculated the mean and maximum water levels across Sirajganj District in the 6 months prior to each participant's survey date. We obtained data on seasonal and permanent surface water levels at 30 meter resolution from the Global Surface Water Explorer dataset [32]. We defined seasonal surface water as an area which is underwater for less than 12 months per year and permanent surface water as water that is present year-round. We calculated the distance from each household to the nearest pixel in which seasonal and permanent surface water was present as well as the proportion of the area around each household that contained seasonal and permanent surface water (radii: 10, 25, 50, 75, 100, 250, 500, 1000m) on the date of survey collection.

## Outcomes

We measured depressive symptoms among pregnant women using the Edinburgh Postnatal Depression Scale (EPDS). This tool has previously been validated for evaluating prenatal depression in rural South Asian settings [33,34]. The EPDS score ranges from 0-30, with higher values indicating more severe depressive symptoms. We administered the validated Bengali version of the tool [35]. The instrument was administered by two enumerators with bachelor's degrees who completed 7 days of training. There was an initial pilot with 25 pregnant women in the icddr,b hospital in Dhaka, followed by a second pilot in Chauhali sub-district. We classified women as moderately or severely depressed for total EPDS scores $> 9.5$ and as severely depressed for total EPDS scores $> 13$ [35]. To assess associations with individual EPDS questions, we created binary indicators. Responses that indicate no or few depressive symptoms were coded as 0 and responses that indicate depressive symptoms for that particular question were coded as 1.

## Statistical analyses

We pre-specified the statistical analysis plan (https://osf.io/vw9bj/). We estimated prevalence ratios for moderate or severe depression, severe depression, depressive symptoms for individual EPDS questions, and mean differences in the EPDS score using generalized linear models with a Gaussian family for the EPDS score and a binomial family for the prevalence of depression. We also fit regression models for the number of days latrines were flooded; there was too little variation in the days of home compound flooding to fit models for this variable. We fit both crude and adjusted models. Potential confounders included the month of survey completion, a wealth index generated using the first principal component of household assets, women's age in years, women's years of education, their spouse's years of education, and gestational age in weeks. We only adjusted for covariates associated with each outcome (likelihood ratio test p-value $< 0.2$). We calculated robust sandwich standard errors to account for clustering within geographic block [36]. To assess potential residual unmeasured confounding, we calculated E-values for adjusted measures of association with flooding [37]. The E-value is the minimum strength of association that must be present between an unmeasured confounder and both the exposure and the outcome to explain away the estimated measures of association [37]. Because the survey was cross-sectional and there were limited missing values, we performed a complete case analysis. Replication scripts are available here: https://github.com/jadebc/flood-depression-public.

PLOS Global Public Health

### Focus group discussions

To understand pathways through which flooding increased the risk of prenatal depression and to identify any further preparatory factors contributing to flooding resilience, we analyzed transcripts of focus group discussions (FGDs) conducted as part of an ongoing study related to climate resilience in rural Bangladesh. Participants were women not part of the CRADLE trial but who resided in Sirajganj and Tangail districts and were exposed to flooding risk. FGDs are a method of collecting qualitative data in a small group of participants, administered by a moderator, with prompts and discussion regarding participants' lived experiences and perceptions [38,39]. Participants were adult (age 18+) females residing in the CRADLE study region. We did not collect information about pregnancy status from participants. Two FGDs were conducted in Bengali with 9–11 participants per group on August 13 and 14, 2024, and discussions lasted for up to one hour. The FGD locations were selected so that one community represented adult women residing on the char (sandbar), while the other represented adult women residing adjacent to the river on the mainland. A trained moderator conducted each FGD and audio-recorded the session, and other research staff took notes to aid in subsequent transcription and analysis. The moderator used a FGD guide developed by the research team in English before translation into Bengali. The guide began with introductions of participants before covering these topics in order: experiences with extreme weather, adaptation strategies, income impacts, displacement and shelters, migration, school enrollment and attendance, health, menstruation during floods, and anticipatory adaptive behaviors. Women were asked about health problems, challenges during floods, and gendered difficulties faced during floods but not about mental health directly.

Audio recordings of the FGDs were transcribed into Bengali and then translated to English for analysis. Two authors then coded the English version of the transcripts using Nvivo 14 [40]. A deductive approach [41] was employed based on the predetermined themes of flood preparedness and flood-related vulnerabilities for women [42]. Intercoder reliability based on percent agreement was 81%.

### Mixed methods triangulation

We triangulated the results of the quantitative cross-sectional survey and the focus group discussions through integrated analysis [43,44]. While the quantitative survey was performed in a large sample and is more representative of the population in the study site, the format of the survey may have not elicited a wide range of flood-related vulnerabilities or adaptation strategies that could elucidate mechanisms for the impact of flooding on depression. On the other hand, focus group discussions can elicit the process through which flooding influences mental health, but its smaller sample size may reduce the generalizability of findings. Because the participants in the quantitative and qualitative aspects of our study did not overlap, we analyzed them separately and then we integrated the results from each, giving them equal weight.

This study was approved by the International Centre for Diarrhoeal Disease Research, Bangladesh (icddr,b) Ethical Review Committee (PR-22069) and the Stanford Institutional Review Board (63990).

### Ethics statement

We obtained written consent from pregnant women who participated in the randomized controlled trial and adult women who participated in the FGDs.

## Results

### Study participant characteristics

We administered the quantitative survey to 881 pregnant women, from September 24, 2023 to July 19, 2024. Overall, characteristics were similar between participants who had vs. had not exposed to flooding, however it appeared that women with no flooding exposure had higher socioeconomic status compared to those who had experienced flooding (S1 Table). Mean years of education was higher among those unexposed vs. exposed (6.3 vs. 5.6); the percentage with access to basic

sanitation was also slightly higher among those unexposed vs. exposed (17.6% vs. 14.1%). More than half of the house-holds (59%) had a monthly income less than USD 100 and the average household size was 5 members. Most homes were constructed with a tin roof and tin wall, and all homes had soil floors because having a soil floor was an enrollment criterion for the parent trial. All households had basic, improved water (defined as drinking water from an improved source from which collection time is no longer than 30 minutes for a roundtrip, including queuing), 71% had unimproved sanitation (based on the type of flush and toilet slab), and 23% had a handwashing station with soap and water.

### Flooding and water exposures

Approximately ten percent of households in the quantitative survey sample reported that their union was flooded for at least one day in the past six months (Table 1, Fig 1). In the past 6 months, the compound was flooded for at least one day

**Table 1. Summary of flooding among survey participants.**

|  | N | %/ mean |
| --- | --- | --- |
| Union flooded for at least one day in past 6 months | 878 | 10.3% |
| Compound flooded for at least one day in past 6 months | 881 | 3.6% |
| Number of months ago that the compound flooded | 32 | 2.3 months |
| Inside of the home flooded in the past 6 months | 881 | 0.30% |
| Number of days the home was flooded | 3 | 14 days |
| Latrine flooded in past 6 months | 881 | 1.2% |
| Number of days latrine was flooded | 11 | 12.5 days |
| Tubewell flooded in past 6 months | 881 | 0.6% |
| Number of days the tubewell was flooded | 5 | 11.4 days |
| Respondent feels prepared to handle a flood if it happened tomorrow | 879 | 28.6% |

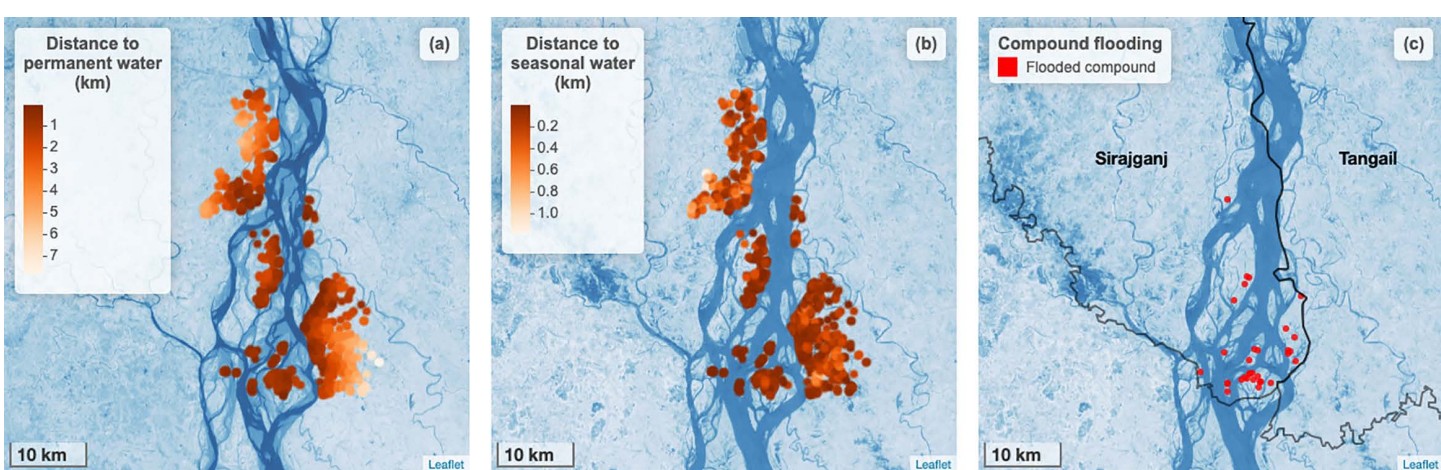

**Fig 1. Surface water proximity and home compound flooding in each study household.** a) Points indicate the kilometers from each study house-hold to permanent surface water. Base map shows permanent water bodies present during the dry season (November 2022 to May 2023). b) Points indicate the kilometers from each study household to seasonal surface water. Base map shows seasonal water bodies present during rainy season (June 2023 to October 2023). c) Red circles indicate households that experienced flooding in their compound in the 6 months prior to the study. Base map shows seasonal water bodies present during rainy season. In all panels, the base map is based on the Landsat 8-Day TOA Reflectance Composite, courtesy of the U.S. Geological Survey. Darker shades of blue indicate a higher likelihood that surface water was present. https://developers.google.com/earth-engine/datasets/catalog/LANDSAT_LC08_C02_T1_TOA.

in 3.6% (n = 32) of households and the average length of flooding was 2.3 days. Three households (0.3%) reported that the inside of their home was flooded during this period and the flood lasted for an average of 14 days. Within the same recall period, flooding affected the tubewell (a type of well where a pipe is bored underground) in 0.6% of homes and the latrine in 1.2% of homes with the mean flooding duration being 12.5 days and 11.4 days, respectively. Rainfall primarily occurred between April and October (Fig 2a). The measured water level was highest between July and September (Fig 2b). During the recall period, women reported flooding events between the months of July and October (Fig 2b). Participant-reported flooding events tended to co-occur with elevated water levels. On average, the proportion of the 1km around each compound that contained permanent surface water was 0.014 (range: 0.000 to 0.199). The proportion of the

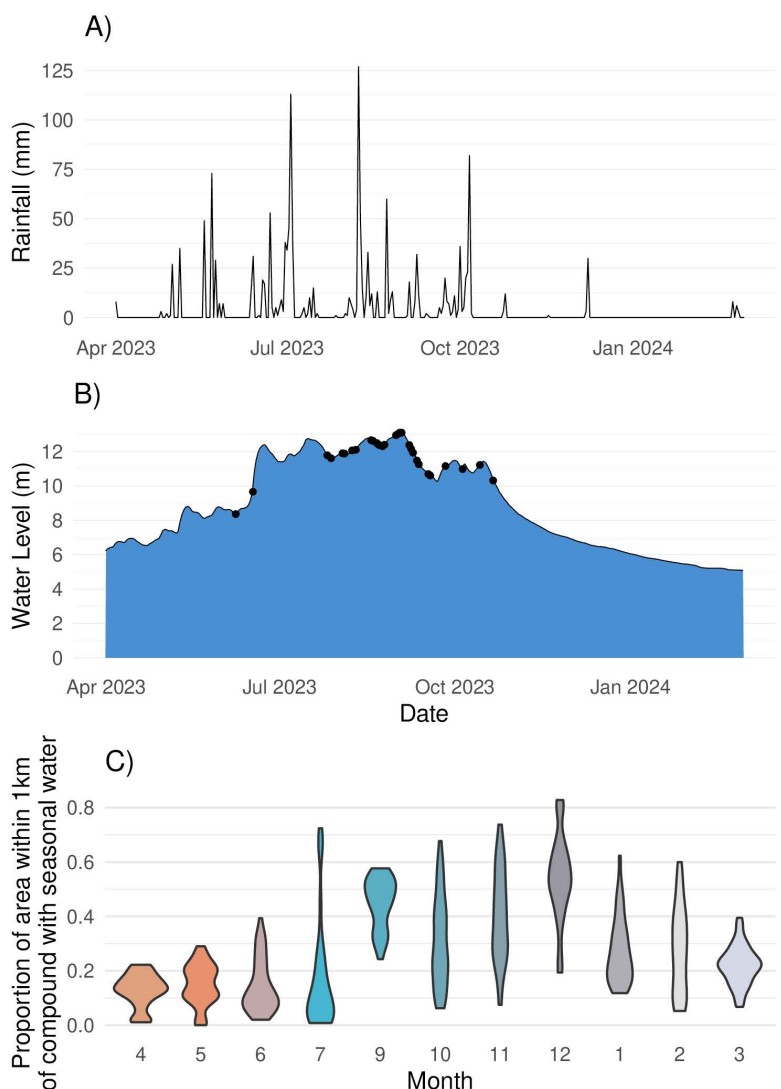

**Fig 2. Rainfall, water level, and seasonal surface water patterns over time.** a) Total daily mm of rainfall in Sirajganj district. b) Water level (meters above sea level) in Sirajganj district by date during the recall period for the study. Black points indicate dates when there was at least one flooding event reported by a participant. c) Violin plots showing the distribution of the proportion of area within 1km of compounds with seasonal surface water. The 1km radius is shown since all study households had seasonal surface water within this radius during the study period. Orange shades indicate the hottest months; teal shades indicate rainy season months; gray shades indicate drier and cooler months.

1km around each compound that contained seasonal surface water varied, with higher values during the rainy season and lower values in April and May, which tend to be hottest (Fig 2c).

All participants from the focus group discussions reported experiencing a severe flood three years ago, during which their homes were flooded. FGD participants residing on chars reported floods occurring every other year, while FGD participants residing on the mainland reported floods occurring almost annually.

*"There was a huge flood in 1988, then in 1994 and in 1998. After 2000, there is flood after flood happening here. There was just no flood for three years in between."* (FGD participant, age 47)

*"Within just one night, the houses become destroyed and taken away in the river by the flood water. If it would have been midnight, even people would not be able to survive as they were asleep."* (FGD participant, age 35)

### Association between flooding and depression

Flooding was associated with higher prevalence of prenatal depression (Table 2). Women who experienced latrine flooding had a 3.58-fold (95% CI 2.22, 5.75) higher prevalence of depression and those who experienced flooding of their home compound had a 2.08-fold (95% CI 1.24, 3.51) higher prevalence of depression compared to those who did not experience either type of flooding. Latrine flooding was also associated with a 4.33-fold (95% CI 1.37, 13.73) higher prevalence of severe depression compared to no latrine flooding. Flooding at the union level was associated with a 1.41-fold (95% CI 0.97, 2.04) higher prevalence of prenatal depression compared to no flooding in the union, but the confidence interval included the null. The duration of flooding in the home compound was not associated with depression or EPDS scores (S3 Table). E-values for depression analyses indicated that unmeasured confounding would have had to be strong to fully explain estimated associations between home compound or latrine flooding and depression (S2 Table).

Prevalence ratios were estimated with generalized linear models with a Poisson family and log link. The dependent variable was depression, and the independent variable was flooding. Adjusted models also included any of the following covariates that were associated with each outcome (likelihood ratio p-value<0.2): month, wealth index, mother's years of education, spouse's years of education, mother's age, gestational age. Robust sandwich standard errors accounted for clustering at the block level.

On average, women who experienced latrine flooding had an EPDS score that was 3.87 points (95% CI 0.72, 7.02) higher compared to those without latrine flooding (S3 Table). Women who experienced home compound flooding had a 2.34 point (95% CI 0.13, 4.55) higher EPDS score than those who did not experience home compound flooding. We did not find evidence of an association between union flooding and the EPDS score.

**Table 2. Association between flooding and depression.**

|  | N | Prevalence among exposed | Prevalence among unexposed | Crude prevalence ratio (95% CI) | Adjusted prevalence ratio (95% CI) |
|---|---|---|---|---|---|
| **Moderate or severe depression** | | | | | |
| Flooded latrine | 881 | 63.6% | 19.7% | 3.24 (2.17, 4.83) | 3.58 (2.22, 5.75) |
| Flooded compound | 881 | 43.8% | 19.3% | 2.26 (1.45, 3.54) | 2.08 (1.24, 3.51) |
| Flooded union | 878 | 27.8% | 19.3% | 1.44 (1.00, 2.07) | 1.41 (0.97, 2.04) |
| **Severe depression** | | | | | |
| Flooded latrine | 881 | 27.3% | 7.2% | 3.77 (1.47, 9.68) | 4.33 (1.37, 13.73) |
| Flooded compound | 881 | 12.5% | 7.3% | 1.71 (0.64, 4.57) | 1.42 (0.49, 4.12) |
| Flooded union | 878 | 8.9% | 7.2% | 1.23 (0.61, 2.46) | 1.19 (0.59, 2.41) |

We also measured the association between individual EPDS questions and home compound flooding (Table 3). Women who experienced home compound flooding had a 2.06-fold (95% CI 1.35, 3.12) higher prevalence of being unable to see the funny side of things and a 1.88-fold (95% CI 1.30, 2.72) higher prevalence of finding it difficult to look forward to enjoyment compared to women who did not experience home compound flooding. Additionally, home compound flooding was associated with a 8.40-fold (95% CI 5.20, 13.35) higher prevalence of experiencing thoughts of self-harm. Other individual EPDS responses were not associated with flooding.

Prevalence ratios were estimated with generalized linear models with a Poisson family and log link. The dependent variable was depression, and the independent variable was flooding. Adjusted models also included any of the following covariates that were associated with each outcome (likelihood ratio p-value<0.2): month, wealth index, mother's years of education, spouse's years of education, mother's age, gestational age. Robust sandwich standard errors accounted for clustering at the block level.

FGDs revealed multiple mechanisms through which flooding influenced women's physical health and safety, and in turn, mental health (Box 1). Women reported having to temporarily relocate during severe floods, staying with relatives, neighbors, or in government shelters. However, women residing on the chars did not have access to government shelters and often had no choice other than to temporarily live on the roofs of their homes.

*"We need a shelter in this village. If there is a flood, we can take shelter there. We may stay for 20-25 days and after the water level goes down, we may get back to our houses from the shelter. But there is no system for shelter."* (FGD participant, age 47)

Participants reported that they had minimal savings available to prepare for a flood. Some participants went into debt during or following floods to purchase materials to reinforce their home or extra food to store during floods. Livestock are an important source of income, and when they drowned during floods, it negatively impacted women's income sources.

Additionally, food insecurity was a challenge for participants during periods of displacement following floods, and some had difficulty obtaining sufficient food while residing in shelters.

*"They should also arrange food supply in that shelter so people will be able to eat. People who have their own safe place can eat in their home and those who do not will be able to eat there. People of the chars have stayed on the roof of their house during floods. We have been there, we have seen and experienced this. They somehow stayed there and if someone provided them chira or puffed rice, they ate it otherwise they remained without food.* (FGD participant, age 45)

**Table 3. Adjusted association between individual EPDS questions and home compound flooding.**

|  | Prevalence among exposed | Prevalence among unexposed | Adjusted prevalence ratio (95% CI) |
|---|---|---|---|
| Difficult to see the funny side of things | 46.9% | 20.3% | 2.06 (1.35, 3.12) |
| Difficult to look forward to enjoyment | 43.8% | 20.0% | 1.88 (1.30, 2.72) |
| Blamed self unnecessarily when things went wrong | 9.4% | 17.1% | 0.50 (0.12, 2.00) |
| Felt anxious or worried for no good reason | 21.9% | 19.9% | 1.09 (0.54, 2.22) |
| Felt scared or panicky for no good reason | 25.0% | 10.5% | 2.37 (1.22, 4.62) |
| Things getting on top of self | 31.2% | 22.6% | 1.26 (0.74, 2.15) |
| Difficulty sleeping due to unhappiness | 15.6% | 19.0% | 0.72 (0.30, 1.73) |
| Felt sad or miserable | 25.0% | 21.8% | 1.02 (0.52, 2.02) |
| Cried due to unhappiness | 28.1% | 13.9% | 1.84 (0.97, 3.50) |
| Thoughts of self-harm | 43.8% | 4.4% | 8.40 (5.29, 13.35) |

Participants also reported experiencing food insecurity due to damaged crops and vegetables. During floods, women avoided going to the market to purchase food until it became absolutely necessary. Women were concerned about leaving their children unattended to go purchase food due to the risk of drowning. Participants also reported reducing their food intake to ensure enough food for their children.

*"We face problem everywhere from cooking to taking care of children. Also, the child may fall into the flood water if we do not always give proper attention. I can't go anywhere leaving the kids at home. We have to leave the children with someone and go to work. So, is this not a problem? We have to stay alert always."* (FGD participant, age 45)

*"We cannot have three meals a day. It is difficult to have at least one meal a day during floods. Even if we can manage one meal a day, we have to give most of the food to our children. Because if we eat according to our hunger, we will not be able to give them enough food. That's why we give them enough food by sacrificing from our sides."* (FGD participant, age 45)

*"I have to take only one meal in a day. We cannot have three meals. Few people have rice in their house and few don't. During floods, it is difficult to move because there are very few boats, no one to swim and also, we don't have money to hire a boat to go to the market. We have to suffer a lot."* (FGD participant, age 45)

---

**Box 1.–Flooding-related vulnerabilities identified in focus groups with adult women**

**Housing security**

- Displacement
- Damage to home and belongings

**Food security**

- Damaged food, unable to travel to market to purchase food due to floodwater or having to stay vigilant to prevent children from drowning
- Cannot operate stove to cook
- Dietary changes due to limited food availability; having to rely on stored dry food
- Damage to food crops for household food consumption and income generation

**Income, savings, and livelihoods**

- No savings available to prepare for a flood
- Livestock/poultry drowning impacts women's livelihoods
- Going into debt due to borrowing for home preparation or purchasing extra food prior to floods

**Health and physical safety**

- Disease (e.g., skin rash, diarrhea)
- Venomous snakes entering the home
- Increased risk of drowning

---

- Domestic violence
- Harassment of girls in shelters

**Water, sanitation, and hygiene**

- Limited drinking water access
- Open defecation
- Using temporary, unhygienic latrine
- Relieving oneself less often
- Lack of privacy when using latrines
- Wading through flood water to access latrines
- Waiting in long queue to use neighbor's latrine
- Soiling oneself while waiting to access a latrine
- Difficult to bathe
- Difficult to maintain menstrual hygiene

All FGD participants reported experiencing domestic violence during floods. Participants stated that violence occurs when there is income loss. Participants reported having violent arguments when they ask their husbands for money to buy food during floods when there is limited income and financial losses.

*"It is the month of floods. There is no income in the family. So, when I ask for something he (my husband) starts to argue with me. If I tell him to buy some groceries, he starts to hit me. So, I do not express myself, usually out of fear of violence. Still, sometimes I have to tell him when there is no way."* (FGD participant, age 30)

The number of discrete vulnerabilities that women experienced were the largest for water, sanitation, and hygiene, with women facing multiple challenges to bathing, maintaining menstrual hygiene, and relieving themselves during floods.

*"When the water level increases, we cannot eat properly to satisfy our hunger because we know that if we eat enough food to satisfy our hunger, we will have to use the washroom frequently. That is why we try to consume less. During the daytime, we go to a house which is not flooded and use their washroom. We have to go there by crossing flood water. Our clothes get wet and after returning home, we have to change it. That is why we try to consume less so that we do not have to use the washroom more than once a day."* (FGD participant, age 47)

In addition to finding it difficult to dry cloths used for menstrual hygiene during floods, one participant reported having to wash the cloths in the river instead of using tubewell water.

*"In usual times, we may use one or two cloths because we can wash it and dry it. But during floods, we have to throw it away. So, we use five to six cloths. We suffer a lot.*" (FGD participant, age 30)

Most women reported finding it difficult to have sufficient privacy when using the toilet, feeling shy about having to use a neighbor's toilet or worrying about facing harassment while doing so, especially because their clothes became wet while

trying to locate a toilet during floods. All participants agreed that there were times when they spent so long looking for a private toilet that they soiled their clothes. They also reported having to construct makeshift toilets when their toilets are flooded.

> *"Women face challenges in using toilets. Women cannot defecate anywhere. They need privacy. When woman goes to the toilet of others, the surrounding people tease them. We feel shy. This is really challenging. People do not make fun of it when men go to the toilet of others. When infants defecate, it can be thrown away outside. Men can do in any where or on the roadside areas. But, women cannot do that without privacy. We feel shy in going to people's house for toilet. So, we go to people's house secretly for using the toilet. Moreover, not so many people can go to the same house. It does not look good."* (FGD participant, age 40)

**Association between surface water proximity, water level and depression**

We did not observe associations between distance to any surface water and the prevalence of prenatal depression (S4 Table). Each additional meter away from permanent surface water was associated with a 0.27 (95% CI 0.10, 0.44) lower mean EPDS score; we did not observe an association with seasonal surface water (S5 Table). Overall, we did not observe associations between the proportion of the area near households that contained surface water and depression (S1 File).

There was a positive association between water level and mean EPDS score in adjusted models. A one-meter increase in mean water level above sea level in the past 6 months was associated with 0.26 (95% CI 0.13, 0.39) higher mean EPDS score. We did not observe associations between water level and the prevalence of depression (adjusted PR = 1.03 (95% CI 0.97, 1.10)) or severe depression (adjusted PR = 1.02 (95% CI 0.91, 1.14)).

In the focus group discussion, one participant reported that homes located close to the river were damaged due to the high water level during floods.

> *"Our house was on the edge of the river bank which was damaged entirely by this year's flood. Those who had their houses on the river bank got damage to their house this year. If the water level would have been lower, there would not be that much damage."* (FGD participant, 20)

**Anticipatory practices related to flooding**

In the quantitative survey, only 28.6% of women reported feeling prepared to handle a flood if it occurred the next day. One of the most common flood preparedness strategies was temporarily moving to a safe location (Table 4). Nearly half of respondents (40.0%) said they would move to higher ground or a dry area during a flood. Other temporary relocation strategies included living on a suspended platform (13.5%), moving to a flood shelter (4.3%), moving to someone else's home (2.3%), living on a boat or raft (2.1%), and living on the road (1.7%). Common preparedness strategies other than relocation included collecting dry food (21.1%), preparing a portable stove (3.5%), and raising the height of the bed (9.2%).

Focus group participants mentioned most flooding adaptation strategies mentioned in the survey, plus five additional strategies not included in survey responses (Table 4). Almost all FGD participants reported storing dry food in preparation for floods. Others reported strategies included preparing a dry stove, constructing temporary suspended platforms, or raising beds so that family members and domestic animals can stay above flood water during floods. Some participants reported saving money in advance to prepare for floods, one participant mentioned taking loans, while others mentioned being unable to save money as preparation. Regarding transportation during floods, participants reported using boats or rafts made of banana trees. Participants living on chars also described storing more pieces of old cloth than usual as preparation for menstruation. This is necessary since it is difficult to air dry and reuse the same piece of cloth when there is flooding.

**Table 4. Anticipatory practices related to floods.**

| | Quantitative survey %(N = 881) | Focus group discussion |
|---|---|---|
| Temporarily relocate to higher ground or dry area | 40.00% | ✓ |
| Collect dry food | 21.10% | ✓ |
| Temporarily relocate to suspended platform | 13.50% | ✓ |
| Raise bed | 9.20% | ✓ |
| Temporarily relocate to shelter | 4.30% | ✓ |
| Prepare stove | 3.50% | ✓ |
| Temporarily relocate to another home | 2.30% | ✓ |
| Temporarily relocate to boat or raft | 2.10% | |
| Temporarily relocate to road | 1.70% | |
| Other | 1.10% | ✓ |
| Raise home | 0.50% | ✓ |
| Prepare cash | 0.40% | ✓ |
| Temporarily relocate to roof | 0.30% | ✓ |
| No preparation | 0.10% | |
| Reinforce home structure (banana trees, bamboo poles, wrap foundation in polythene to prevent erosion) | 0.00% | ✓ |
| Use or borrow a boat (e.g., to travel to buy food) | 0.00% | ✓ |
| Move livestock to raised ground or raised platform | 0.00% | ✓ |
| Treat drinking water | 0.00% | ✓ |
| Receive flood warning text messages from non-governmental organizations | 0.00% | ✓ |

A majority of FGD participants living on the mainland reported that the inside of their homes no longer floods during annual floods because they have raised the foundation of their homes.

*When our previous houses got damaged, we made an elevated house with soil here. After that, we have not faced any flooding in house. We just have to walk through the flood water when we need to go to the toilet.* (FGD participant, age 45)

On the other hand, one participant living on the char reported not raising their home because they anticipated frequent damage from floods. Frequent damage to homes is likely expected in this setting since all FGD participants living on chars reported having moved their home structures 5-6 since their marriage.

*After working the entire day in the field, I had to repair the house in the night. Sometimes, it takes almost ten years to build a strong house. But, it will be of no use because one extreme flood can take the entire house in the river. That's why we live in a hut, we do not add anything to make it stronger.* (FGD participant, age 40)

Participants also mentioned constructing suspended platforms or elevating their beds to protect their livestock and children.

*"If we have chicken, we plan to prepare a "machal" (elevated platform) for the chickens so that they can sit on it. For the children, we decide to raise our bed so that the water does not reach there."* (FGD participant, age 47)

However, most participants living on the mainland reported elevating the plinth of their homes. Other home modification strategies were reinforcing the plinth with banana trees and branches, covering the plinth with polythene, and planting trees around the home to prevent damage to the plinth during floods. Most participants reported discussing with family members about relocation to relatives' or neighbors' homes that are located on higher ground during floods. Some participants reported relocating to government shelters if necessary.

## Discussion

Flooding of compounds and latrines was associated with higher prevalence of prenatal depression and thoughts of self-harm in rural riverine communities in Bangladesh. Latrine flooding had a particularly strong association with prenatal depression. Higher water levels and shorter distances to permanent surface water were associated with higher EPDS scores. Focus group discussions substantiated these findings and revealed the multitude of ways in which flooding events influenced mental health by, for example, preventing pregnant women from bathing and relieving themselves. We also identified multiple ways that some women prepared for and/or remained resilient during flooding events, including home modifications, storing food, saving money, and treating drinking water.

Overall, the frequency of flooding and prevalence of depression in our study population was within the range of estimates in other studies in rural Bangladesh [45,46]. Compared to a prior study in Bangladesh investigating flooding and depression in the general population, we found stronger associations with prenatal depression [28]. This is consistent with our expectation given that women in Bangladesh are more negatively impacted by flooding than men [47]. Pregnant women could be more vulnerable to flooding because normal changes to the brain during pregnancy could interrupt necessary responses to floods, and floods could directly result in brain rewiring during pregnancy [30]. We found higher levels of suicidal ideation following floods than a prior study did in a flood-prone area of Bangladesh [33]; this difference could reflect an acute response to a flooding event as opposed to a persistent flood risk.

Our quantitative and qualitative analyses both pointed to inadequate sanitation and hygiene during floods as a potentially important driver of prenatal depression. In the quantitative survey, latrine flooding was associated with 3.6-fold higher moderate or severe depression and 4.3-fold higher severe depression. In FGDs, women mentioned multiple vulnerabilities related to sanitation and hygiene but did not report any adaptation strategies in either the quantitative survey or FGDs (Box 1). During floods, women faced many barriers to bathing, maintaining menstrual hygiene, or relieving themselves; we expect that these particular vulnerabilities would be exacerbated during pregnancy and could be key drivers of prenatal depression. A prior review identified multiple physical, financial, and social stressors resulting from a lack of access to sanitation, with stronger psychosocial impacts on women [48]. Safe sanitation and hygiene is essential to women of childbearing age to maintain hygiene during menstruation, spotting during pregnancy, and bleeding for 4–6 weeks following miscarriage and postpartum [49]. Our focus group findings were consistent with reports in prior studies that women experience shame and risk of violence when they must go outside the home setting to access safe water or sanitation, especially if their clothing is wet [13,50]. Upgrading water, sanitation, and hygiene infrastructure to be climate resilient may be necessary to sustain women's mental health during floods.

FGDs also revealed other potential pathways through which home compound flooding influenced women's mental health, including displacement, food insecurity, threats to physical safety, children's risk of drowning, gender-based violence, additional responsibilities in the home, and economic hardship, consistent with prior studies [51]. Prior to floods, women in rural Bangladesh are traditionally responsible for preparing portable stoves and raising the height of the home or bed [52]; following floods, it may be more difficult to keep up with household responsibilities due to damaged or lost household assets [44]. Women reported that they were less likely to eat during floods, consistent with prior studies [47]. Major floods are associated with food insecurity in Bangladesh, with women who experienced a major flood having higher odds of depression even 2.5 years after the event [53]. Gender norms, including purdah, which requires women to remain within the compound unless accompanied by a male relative, can also prevent women from taking actions to prepare for

or safely relocate during floods [22]. Climate-resilient household construction that prevents flood water from entering the home and latrine and reduces the risk of temporary displacement may help reduce gender inequities in mental health following flooding.

The adaptation strategies households used to prepare for and respond to floods were similar to those reported in other studies in Bangladesh. These included temporary relocation to a shelter, raised platform, or boat; raising their home or reinforcing it; storing food; and saving cash. Notably, participants of both the quantitative survey and FGDs mentioned multiple adaptation strategies related to altering their homes during flooding events to reduce the chance of displacement due to damage to their homes or belongings. However, none mentioned preparatory upgrades to their water, sanitation, or hygiene infrastructure. A study conducted in remote chars, or sandbars, identified adaptation strategies such as treating drinking water, storing food, finishing work early in the morning, selling assets, and taking out loans [51]. Additional flooding adaptation practices in Bangladesh include floating agriculture and raised houses, tubewells, and latrines [54]. However, overall, adaptive capacity remains low: it is estimated that one third of households in northern Bangladesh have the capacity to recover from floods through adaptations such as altered agricultural practices [55]. At the country level, Bangladesh ranks low in adaptation, with an ND-GAIN score of 27 in 2022 (maximum score 100) for its readiness to effectively invest in climate adaptations [56]. Our findings suggest that the adaptation strategies in use in this population may be insufficient for climate resilience and that it would be valuable to integrate depression screening and mental health support into climate adaptation strategies. Screening and mental health support may be implemented during antenatal check-ups at government health facilities or through interventions by non-government organizations (NGOs).

A strength of our study is our mixed methods approach, which allowed us to quantify associations with flooding and to capture contextual and cultural factors contributing to depression following floods, consistent with recommended practices for research on climate and health [57]. This study was subject to limitations. Estimated associations between flooding and prenatal depression from the quantitative survey may be subject to residual confounding since we did not have data on potential confounders such as pre-pregnancy depression, other mental health conditions, antenatal care, pregnancy symptoms, domestic violence, or social support. However, our E-value analyses showed that confounders would have had to be very strong to explain away associations with home compound and latrine flooding. Also, self-reported flooding may be subject to recall error or courtesy bias; it is possible that women who were depressed may have been more likely to recall flooding, which would have biased estimates away from the null. However, a strength of our study is that we also included objectively measured exposures (distance to surface water, water level); scientific inferences across exposures were similar for analyses of depression scores. While our primary exposure variable was self-reported flooding in the prior 6 months, households located closer to water bodies may have experienced more floods prior to this recall period; flooding may have had cumulative effects on mental health. If so, it is possible that our estimated associations between flooding in the prior 6 months and current depression were over-estimated. An additional limitation is that we did not collect pregnancy status information from focus group participants, so qualitative findings might not fully generalize to the results of the cross-sectional survey of pregnant women. We also did not directly ask about mental health in the focus group discussions, although most discussions centered around the stressful consequences of flooding. This internal consistency between analyses with different exposures supports the validity of our inferences. Finally, our cross-sectional design only allowed us to quantify associations between flooding and prenatal depression; longitudinal studies including causal mediation analyses are needed to establish causal relationships and potential causal pathways.

In conclusion, our findings highlight the need to prioritize prenatal depression as a significant public health concern in flood-prone regions of Bangladesh, especially in the context of increasing climate-related hazards. Though study participants report multiple adaptation strategies, strong associations with depression suggest that current strategies are insufficient for climate resilience. Integrating depression screening into antenatal care services in flood-prone regions and among women who reside close to surface water, building flood-resilient homes and latrines, and post-flood mental health support may improve mental health of pregnant women living in vulnerable areas.

## Supporting information

**S1 Table. Characteristics and flood experiences of participants.**
(DOCX)

**S2 Table. E-values for prevalence ratios with depression.**
(DOCX)

**S3 Table. Association between flooding and EPDS score.**
(DOCX)

**S4 Table. Association between distance to surface water and moderate/severe depression and severe depression.**
(DOCX)

**S5 Table. Association between distance to surface water and EPDS score.**
(DOCX)

**S1 File. Association between the proportion of area near households that contained surface water and depression.**
(CSV)

**S2 File. FGD guidelines.**
(DOCX)

## Author contributions

**Conceptualization:** Suhi Hanif, Farjana Jahan, Liza Goldberg, Natalie Herbert, Reza Mostary Akhter, Mahbubur Rahman, Fahmida Tofail, Gabrielle Wong-Parodi, Jade Benjamin-Chung.

**Data curation:** Jannat-E-Tajreen Momo, Farjana Jahan, Liza Goldberg, Natalie Herbert, Afsana Yeamin, Reza Mostary Akhter, Sajal Kumar Roy, Gabrielle Wong-Parodi.

**Formal analysis:** Suhi Hanif, Jannat-E-Tajreen Momo, Jade Benjamin-Chung.

**Funding acquisition:** Jade Benjamin-Chung.

**Investigation:** Jade Benjamin-Chung.

**Methodology:** Suhi Hanif, Jannat-E-Tajreen Momo, Farjana Jahan, Liza Goldberg, Natalie Herbert, Afsana Yeamin, Abul Kasham Shoab, Reza Mostary Akhter, Gabriella Barratt Heitmann, Ayse Ercumen, Mahbubur Rahman, Fahmida Tofail, Gabrielle Wong-Parodi, Jade Benjamin-Chung.

**Supervision:** Jade Benjamin-Chung.

**Visualization:** Suhi Hanif, Gabriella Barratt Heitmann, Jade Benjamin-Chung.

**Writing – original draft:** Suhi Hanif, Jade Benjamin-Chung.

**Writing – review & editing:** Jannat-E-Tajreen Momo, Farjana Jahan, Liza Goldberg, Natalie Herbert, Afsana Yeamin, Abul Kasham Shoab, Reza Mostary Akhter, Sajal Kumar Roy, Gabriella Barratt Heitmann, Ayse Ercumen, Mahbubur Rahman, Fahmida Tofail, Gabrielle Wong-Parodi.

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
