## [Decision Letter · Decision Letter 0]

PGPH-D-25-00092

Flooding and elevated prenatal depression in a climate-sensitive community in rural Bangladesh: a mixed methods study

Dear Dr Jade Benjamin-Chung

Thank you for submitting your manuscript to PLOS Global Public Health. After careful consideration, we feel that it has merit but does not fully meet PLOS Global Public Health’s publication criteria as it currently stands. Therefore, we invite you to submit a revised version of the manuscript that addresses the points raised during the review process.

We have completed the peer review process and refer you to the enclosed five reviewer comments. While reviewers qualified their comments with the paper needing minor to major revisions, we encourage the authors to address all reviewer comments. Moreover, while all reviewers provide insightful comments, reviewers 4 and 5 provide more comprehensive comments that the authors are encouraged to address. 

Please submit your revised manuscript by14th April 2025. If you will need more time than this to complete your revisions, please reply to this message or contact the journal office at globalpubhealth@plos.org. Please include the following items when submitting your revised manuscript:

We look forward to receiving your revised manuscript.

Kind regards,

Danish Ahmad, MBBS,MSc,MNAMS,PhD,IP-FPH(UK),FRCP(Edin),FRCP(Lon)

Academic Editor

Journal Requirements:

2. Please provide separate figure files in .tif or .eps format.

Additional Editor Comments (if provided):

Dear Authors,

Thank you for the submission.

Please review the below academic editor comments and kindly address them.

1. The introduction provides an overall summary but needs strengthening to focus more on flooding and mental health-related events. As bushfires, earthquakes, and other natural disasters can more severely hamper health service delivery, the authors should focus on the literature on flooding’s impact on the health system and health outcomes. Women/families who experience flooding may also be more income-vulnerable and have other vulnerabilities that increase their baseline risk of prenatal depression; please contextualize that for the study setting of Bangladesh.

2. The links between flooding mental health and climate change aren’t clearly and coherently presented; please work on the reviewer's comments to clarify the rationale and provide a stronger synthesis of the literature to frame the study. Please clearly distinguish between floods and natural disasters as a generic term.

3. Provide added context for the study site (Sirajganj and Tangail districts) and how/why they were selected. Overall, the methods should also provide a research framework that explains why a mixed-method study was considered suitable for the study aims. Additionally, provide details of the randomisation process in the original trial and comment on how the randomisation was done. Ideally, using a difference in difference method to compare intervention (exposure areas) with comparable control areas would have provided more insightful results.

4. Please revise Table S1 to give a descriptive analysis of the exposed versus unexposed and comment on their comparability

Reviewers' comments:

Reviewer's Responses to Questions

**Comments to the Author**

1. Does this manuscript meet PLOS Global Public Health’s publication criteria?

Reviewer #1: Yes

Reviewer #2: Partly

Reviewer #3: Yes

Reviewer #4: Partly

Reviewer #5: Yes

2. Has the statistical analysis been performed appropriately and rigorously?

Reviewer #1: I don't know

Reviewer #2: No

Reviewer #3: Yes

Reviewer #4: Yes

Reviewer #5: Yes

3. Have the authors made all data underlying the findings in their manuscript fully available (please refer to the Data Availability Statement at the start of the manuscript PDF file)?

Reviewer #1: Yes

Reviewer #2: No

Reviewer #3: Yes

Reviewer #4: Yes

Reviewer #5: Yes

4. Is the manuscript presented in an intelligible fashion and written in standard English?

Reviewer #1: Yes

Reviewer #2: No

Reviewer #3: Yes

Reviewer #4: Yes

Reviewer #5: Yes

Reviewer #1: The introduction effectively sets the stage for the study by emphasizing the growing concern of climate change and its impact on mental health, particularly in low- and middle-income countries (LMICs). It provides a strong rationale for focusing on prenatal depression, given its long-term adverse effects on child health and development.

The literature review is thorough, citing relevant studies on the mental health impacts of flooding and the lack of research on prenatal depression in LMICs. The introduction also highlights the unique vulnerabilities of pregnant women in flood-prone areas, such as gender norms and inadequate sanitation, which are critical to understanding the study's context. The introduction could have further emphasized the novelty of the study, particularly its focus on a climate-sensitive community in rural Bangladesh, which is a region highly vulnerable to climate change.

The study employs a mixed-methods approach, combining a cross-sectional survey with focus group discussions. This design is appropriate for capturing both quantitative associations and qualitative insights into the lived experiences of pregnant women in flood-prone areas. The study population consists of 881 pregnant women from riverine communities in rural Bangladesh, a region highly susceptible to flooding. The inclusion criteria are well-defined, focusing on women in their second or third trimester of pregnancy, which is appropriate given the study's focus on prenatal depression. The use of the Edinburgh Postnatal Depression Scale (EPDS) to measure depressive symptoms is appropriate, as it has been validated in similar settings. The inclusion of both self-reported flooding and objective measures of water levels and proximity to surface water strengthens the study's findings.

The statistical methods, including generalized linear models and log-linear models, are appropriate for analyzing the associations between flooding and depression. The use of E-values to assess potential unmeasured confounding is a strength, as it provides a measure of the robustness of the findings. The qualitative component adds depth to the study by identifying key drivers of depression, such as domestic violence, inadequate sanitation, and food insecurity. The thematic analysis approach is well-suited for capturing the nuanced experiences of women in flood-prone areas.

The results are presented clearly, with tables and figures that effectively summarize the key findings. The study finds strong associations between flooding (both compound and latrine flooding) and elevated prenatal depression, as well as thoughts of self-harm additionally with findings which are supported by both quantitative and qualitative data.

The focus group discussions reveal important contextual factors, such as the challenges women face in maintaining hygiene and accessing latrines during floods, which contribute to depression. The qualitative data also highlight the limited preparedness of women for flooding events, despite some adaptation strategies like storing food and home modifications. The results are internally consistent, with both self-reported and objective measures of flooding showing similar associations with depression. This consistency strengthens the validity of the findings.

The discussion could have further explored the implications of the findings for policy and practice, particularly in terms of integrating mental health support into climate adaptation strategies.

Unfortunately, the cross-sectional design limits the ability to establish causality between flooding and prenatal depression and the study relies on self-reported data for some measures, which may be subject to recall bias. In the same vein, the lack of data on pre-pregnancy depression and other potential confounders (e.g., domestic violence, social support) may limit the ability to fully account for all factors influencing depression.

Reviewer #2: 1. The author should work on grammar flow, starting with the background of the abstract down to the entire document; to improve readability.

2. The author should consider being consistent with the information they present. For example, paragram 2 of the abstract says "We fit generalized linear and log-linear models", however the use of GLM in the results is not apparent.

3. Considering this is a mixed methods study; the presentation of results for both qualitative and quantitative results leaves much to be desired. No proper triangulation of the qualitative and quantitative results. qualitative results have no narration (no apparent story to tell, no context). Labeling of tables in the results section is unclear and hard to follow.

4. Study title implies a link between flooding and climate change. However, no clear link is made in the results to speak to the study topic. No climate change modeling was done. Perhaps the author could more specific to what exactly the results speak to and not quickly generalise. Consider building a link that a reader can follow and easily get the message from the text.

5. The study identified several drivers such as domestic violence, food insecurity, etc. However, it does not specify if these drivers mediate the relationship between floods and depression, how are these drivers affecting depression? knowing how these drivers affect depression could be useful for readers that are interested in communing up with interventions. You may need to consider doing some mediation analysis.

6. The presentation of the ethical considerations for this study leaves much to be desired. Did you get ethical clearance or weaver from IRB? Please check the Journal guidelines on what needs to be cleared stated for this sections.

Reviewer #3: Overall impression: insightful and resourceful study that explores an ever more prescient issue not only for Bangladesh but other regions where poverty, poor infrastructure, and flood-related climate change events intersect. I appreciate the availability of all the data and coding in order to replicate the results, and to help other researchers conduct similar studies--supporting collaborative international work is paramount in response to climate change.

My only cause for concern: the findings from the FDGs about domestic violence and social ostracization for having to use someone else's latrine during flooding events in the home. Purdah--and the geosocial context of a poor region in a Muslim country--is barely mentioned, and I believe this needs to be emphasized more strongly throughout the paper. While the authors meet the publishing criteria #4: "Conclusions are presented in an appropriate fashion and are supported by the data," I am concerned about the lack of data collection around gender-based violence (GBV) especially given the geosocial context of this study, and the lacking discussion of GBV (if it wasn't feasible to collect this data). I would strongly encourage the authors to explicitly discuss the realities of GBV and climate change as they relate to this study, including the minor mention of domestic violence hidden in the FGD findings, with more emphasis on the sociocultural influence of purdah and stress related to having to leave the home due to flooding, and the probable unmeasured confounding of GBV on prenatal depression in this study. While the e-value is interesting and informative, if there was a factor that could be strong enough to explain some of the associations between flooding and prenatal depression, I think domestic violence would be a good candidate. I think this added context would greatly improve the impact of the paper, especially for readers outside of this social/cultural reality. I don't think I would be the only reader to wonder about this gap in this discussion about GBV and depression among women, in an otherwise well executed and informative study.

Reviewer #4: The manuscript presents a crucial and rising issue of climate change which is directly and indirectly related to public health concern. Mental health and its importance is further elaborated in the work. Authors have highlighted the vulnerability in the moments of natural disaster like flood which is further aggravated by the climate change impacts. In my review i have following comments and suggestions to improve the article:

In general i request authors to have proof read the document after addressing the review queries appropriately.

It will be better not to use numbers and digits while starting the new paragraph and sentence. Please review and address over the manuscript. The citations are not properly formatted over the manuscript, please review and make them uniform.

Abstract

Ambiguous terminology used as compounds flooded or compound flooding.

Conclusion of abstract is not well framed but only presented the findings please rewrite them.

Introduction section

Setting of background on the study topic would be better in the preliminary section. Clear indication of the study objectives will be better in the introduction section, though authors have indicated the expected outcome.

Methodology

Who and how were FGD participants selected? Were they prenatal group female? If not how do you present the relevance of selecting females out of the prenatal group in this study and their lived experiences associated sith the prenatal group of women?

Were there any specific mixed methods study designs followed? might be for triangulation? How does the design supports the strengths of this study as claimed by the authors? Also explain in brief any triangulation conducted if any and not presented in the article. I advise to present the triangulation throughout the study for maintaining the scientific rigor of the claimed results and conclusions from the study.

Results

The results are well explained and presented but will be better if appropriate triangulation conducted for claiming the findings.

The CRADLE trial mentions 850 as their samples but in the current study the results are presented from 881 pregnant women. Could you explain.

The FGD seems to have been conducted with women only but another query is how the Box 1 could present FGDs with adult men and women.

Additionally; were there any queries asked during FGDs related to health and physical safety which are directly related to prenatal depression and mental health. If yes; how were the responses triangulated with the quantitative findings on Post Natal Depression and flooding. Please present these triangulations as well.

Discussions

The presentation is well along the risk, vulnerabilities associated with climate change and natural disasters among different locations of Bangladesh. These are well discussed with the concurrent findings from national and international evidences but the chief issue of study association of compound flooding and prenatal depression seems behind. So, it should be further discussed with appropriate triangulations over the section.

Another query is: Were there any queries during FGDs related to health and safety? How were the responses triangulated with the quantitative findings of flooding associated with prenatal depression.

Reviewer #5: Referee report on “Flooding and elevated prenatal depression in a climate-sensitive community in rural Bangladesh: a mixed methods study” (PGPH-D-25-00092)

The paper aims to examine the associational impact of floods on the mental health outcomes, as measured by depression, amongst pregnant women in rural Bangladesh. In principle, the paper uses a simple linear regression model to examine the likelihood of prenatal depression amongst those women who witnessed a flood incident in the past 6 months in comparison to those who did not. The author(s) report elevated depression levels for the exposed women. The effect are particularly higher if the women experienced latrine flooding.

The paper highlights an important issue relating to prenatal depression and the author(s) efforts are appreciable. I have a few suggestions for the author(s) to consider.

Major Comments

1. Empirical Strategy:

(a) While authors state that they are using generalized linear model with Gaussian distribution, as per my understanding, this in essence boils down to a simple linear regression model. I urge the author(s) to remove such redundancy and provide more clarity in this context.

Relatedly, author(s) may consider explicitly writing the regression equation in interest of readers.

(b) The empirical strategy compares difference in outcomes of women witnessing a flood event with the women those did not witness such an event in the last 6 months. While flood is a natural disaster (implying it is exogeneous) -- however there are reasons why flooding may an endogenous regressor.

The reason is that the regions (unions) who witnessed a flood in last 6 months may actually be more likely to witness a flood in comparison to regions which did not witness a flood. This is evident in Figure 2b and 2c. Households that are nearer to seasonal water streams are more likely to report compound (and, perhaps latrine) flooding.

If the fact that the regions (unions) who witnessed a flood in last 6 months may actually be more likely to witness a flood in comparison to regions which did not witness a flood, then there is a strong selection bias. In other words, the affected (treated) mothers are more likely to be facing such floods for a continuously more period of time. Thus there remains a possibility that the estimates which the empirical strategy provides may actually be over estimates.

The reason is that floods may induce a perpetual/continued depression -- that may continue beyond 6 months (which is the survey recall horizon). If such affected women (in regions more likely to witness floods) have been suffering from depression for long, then the estimates provided in the table actually represent combining impact of regular flooding rather than floods in the past 6 months.

Relatedly, there may be confounders such as:

o Pre-existing mental health conditions:

o Women who already suffer from anxiety or depression might perceive and report flooding as more traumatic.

(c) Since authors can potentially know the date when the flood actually occurred (using their data), author(s) can see whether the length of exposure to flood also affects the extent of depression. For instance, are those who suffered a flood just one month prior to the survey more/less likely to be depressed than those who suffered 3 months ago. Related to (b), this may add more insights about the temporal impacts of floods.

(d) · The self-reported flooding data may be prone to recall bias, yet there is no discussion of how this could impact the results.

(e) Clustering of standard errors: Since the outcomes within regions (unions) may be highly correlated, author(s) may want to cluster their standard errors at the regio(union) level.

2. Introduction Section

The author(s) may consider adding two additional paragraphs in the Introduction section. One paragraph for adding the key results, and another additional paragraph for adding the highlighting the key contribution of the paper.

3. Descriptive Statistics

Author(s) may consider separating the descriptive statistics from the main results. A separate section for descriptive statistics may be added.

Relatedly, Table 1 is a little confusing. While I understand N=32, N=3, N=11 (etc) represents certain subsamples. However, their corresponding numbers in column (2) are not percentages. Author(s) may consider revising the table to avoid any source of confusion.

Minor Comments

1. Table notes

The author(s) may consider providing detailed table notes describing in detail the outcome variables, the independent variable, and the controls along with a brief about methodology (if deemed necessary). The idea is that the reader should be able to understand the paper just by glancing through the tables even without looking at the text. (This is also just a suggestion)

**Do you want your identity to be public for this peer review?** For information about this choice, including consent withdrawal, please see our Privacy Policy

Reviewer #1: **Yes: ** Benson Tarisai Gombe

Reviewer #2: **Yes: ** Bristol Moonga Ntebeka

Reviewer #3: No

Reviewer #4: **Yes: ** Rabindra Bhandari

Reviewer #5: No

---

## [Decision Letter · Decision Letter 1]

Flooding and elevated prenatal depression in rural Bangladesh: a mixed methods study

PGPH-D-25-00092R1

Dear Dr. Benjamin-Chung,

We are pleased to inform you that your manuscript 'Flooding and elevated prenatal depression in rural Bangladesh: a mixed methods study' has been provisionally accepted for publication in PLOS Global Public Health.

Best regards,

Julia Robinson

Executive Editor

Reviewer Comments (if any, and for reference):

Reviewer's Responses to Questions

**Comments to the Author**

Reviewer #1: All comments have been addressed

Reviewer #2: All comments have been addressed

Reviewer #3: All comments have been addressed

Reviewer #4: All comments have been addressed

Reviewer #5: All comments have been addressed

publication criteria?

Reviewer #1: Yes

Reviewer #2: Yes

Reviewer #3: Yes

Reviewer #4: Partly

Reviewer #5: Partly

3. Has the statistical analysis been performed appropriately and rigorously?

Reviewer #1: Yes

Reviewer #2: Yes

Reviewer #3: Yes

Reviewer #4: Yes

Reviewer #5: Yes

4. Have the authors made all data underlying the findings in their manuscript fully available (please refer to the Data Availability Statement at the start of the manuscript PDF file)?

Reviewer #1: Yes

Reviewer #2: Yes

Reviewer #3: Yes

Reviewer #4: Yes

Reviewer #5: (No Response)

5. Is the manuscript presented in an intelligible fashion and written in standard English?

Reviewer #1: Yes

Reviewer #2: Yes

Reviewer #3: Yes

Reviewer #4: Yes

Reviewer #5: Yes

Reviewer #1: Concisely summarizes objectives, mixed-methods design, key results (e.g., latrine flooding linked to 3.58× higher depression prevalence), and policy implications (sanitation infrastructure).

Limited mention of limitations (e.g., cross-sectional design) and minimal emphasis on qualitative insights (e.g., domestic violence).

Effectively establishes flooding as a climate-sensitive stressor and highlights the lack of LMIC-focused prenatal mental health research.

Gaps Addressed e.g Identifies pathways (e.g., sanitation disruptions, gender norms) linking floods to depression.

Improvements Needed identified the link between climate change projections and study timing (2023–2024 data) could be clearer.

Cross-sectional survey from the CRADLE trial baseline provides a large sample (*n* = 881) but limits causal inference. The mixed-methods approach (quantitative + FGDs) strengthens validity through triangulation.

Statistical Analysis - Adjusted models control for wealth, education, and gestational age. E-values address unmeasured confounding.

- Clustering at the block level accounts for geographic correlation.

- FGDs with 20 women provide rich narratives but lack pregnancy status data, limiting generalizability.

First LMIC study linking prenatal depression to flooding, emphasizing climate-sensitive health risks also highlights sanitation as a critical mediator, informing adaptation strategies (e.g., flood-resilient latrines).

The study provides compelling evidence that flooding exacerbates prenatal depression in rural Bangladesh, driven by sanitation challenges and gendered vulnerabilities. While methodological constraints exist, the mixed-methods approach offers actionable insights for climate adaptation (e.g., resilient infrastructure) and mental health integration into maternal care. Future longitudinal studies could clarify causal pathways and cumulative flood impacts.

Recommendations:

- Prioritize flood-resilient WASH infrastructure in climate adaptation policies.

- Incorporate mental health support into antenatal services in flood-prone regions.

- Expand research on gendered impacts of climate stressors in LMICs.

Reviewer #2: Comments have been addressed. The Authors may consider checking minor punctuation issues to make sure the write-up is flawless.

Reviewer #3: No further comments, the authors adequately addressed my concerns.

Reviewer #4: All comments addressed satisfactorily

Reviewer #5: Regarding data availability: Author(s) have mentioned in the manuscript that "Data is published in Open Science Framework, DOI 10.17605/OSF.IO/VW9BJ".

I am unable to verify, I request the editorial team to verify the same.

**Do you want your identity to be public for this peer review?** For information about this choice, including consent withdrawal, please see our Privacy Policy

Reviewer #1: No

Reviewer #2: **Yes: ** Bristol Moonga Ntebeka

Reviewer #3: No

Reviewer #4: **Yes: ** Rabindra Bhandari

Reviewer #5: No
